# Soil Microbial and Metabolomic Shifts Induced by Phosphate-Solubilizing Bacterial Inoculation in *Torreya grandis* Seedlings

**DOI:** 10.3390/plants13223209

**Published:** 2024-11-15

**Authors:** Yi Li, Yuanyuan Guan, Zhengchu Jiang, Qiandan Xie, Qi Wang, Chenliang Yu, Weiwu Yu

**Affiliations:** 1State Key Laboratory of Subtropical Silviculture, Zhejiang A&F University, Hangzhou 311300, China; lyddeee@163.com (Y.L.); 17816057745@139.com (Y.G.); jiangzhengchu@stu.zafu.edu.cn (Z.J.); xqd2206@stu.zafu.edu.cn (Q.X.); 2School of Forestry and Biotechnology, Zhejiang A&F University, Hangzhou 311300, China; wqqiwang@163.com; 3NFGA Engineering Research Center for Torreya grandis ‘Merrillii’, Zhejiang A&F University, Hangzhou 311300, China

**Keywords:** phosphorus, amplicon sequencing, untargeted metabolomics, biofertilizer, *Torreya grandis*

## Abstract

Phosphorus is crucial for plant growth and development, but excess fertilizer not absorbed by plants often binds with metal ions like iron and manganese, forming insoluble compounds that contribute to soil environmental pollution. This study investigates the impact of *Burkholderia* sp., a phosphate-solubilizing bacterium utilized as a biofertilizer, on the fertility of *T. grandis* soil, alongside the associated shifts in soil metabolites and their relationship with microbial communities after inoculation. The soil microbial community structures and metabolite profiles were analyzed via amplicon sequencing and high-resolution untargeted metabolomics. The inoculation of phosphate-solubilizing bacteria led to a significant (*p* < 0.05) enhancement in total phosphorus, potassium, and nitrogen concentrations in the soil, with a marked increase in available phosphorus in bulk soil (*p* < 0.05). Moreover, the microbial community structure exhibited significant shifts, particularly in the abundance of bacterial phyla such as Acidobacteria, Chloroflexi, Proteobacteria, and the fungal phylum Ascomycota. Metabolomic analysis revealed distinct metabolites, including fatty acids, hormones, amino acids, and drug-related compounds. Key microbial taxa such as Chloroflexi, Proteobacteria, Acidobacteria, Verrucomicrobia, Mucoromycota, and Ascomycota indirectly contributed to soil phosphorus metabolism by influencing these differential metabolites. In conclusion, the application of phosphate-solubilizing bacteria offers an innovative approach to improving soil quality in *T. grandis*, promoting phosphorus utilization efficiency, and enhancing soil ecosystem health by optimizing microbial communities and metabolite compositions.

## 1. Introduction

*Torreya grandis*, a member of the *Taxaceae* family and *Torreya* genus [1], is valued for its high-quality seeds, which are renowned for their distinct flavor and richness in various nutrients and bioactive compounds such as alkaloids, flavonoids, and totarol, giving it considerable economic significance [2]. Phosphorus, an essential element of nucleic acids and membrane lipids [3], is crucial for plant growth and development, and a phosphorus deficiency in soil can hinder these processes [4]. While phosphorus fertilizers can boost plant growth, excess fertilizer not absorbed by plants often binds with metal ions like iron and manganese, forming insoluble compounds that contribute to environmental pollution [5]. In recent years, excessive and unregulated fertilizer application by farmers has caused a severe imbalance in the nitrogen, phosphorus, and potassium ratios in *T. grandis* plantation soils [6]. Consequently, optimizing phosphorus fertilizer use is critical for the sustainable development of *T. grandis*.

Recent research has increasingly focused on understanding how soil microbial diversity and community structures influence soil quality and functional dynamics, representing a key frontier in soil microbiology and agricultural ecology. Rhizosphere microbial communities can affect plant gene expression, thereby influencing plant productivity, soil health, and broader ecosystem functions [7]. The rhizosphere is a critical zone for material exchange between plant roots and soil microorganisms [8,9]. Root exudates from plant metabolism provide nutrients to rhizosphere microorganisms, thus influencing their population and diversity [10]; in turn, the metabolic activities of roots and soil microorganisms impact soil metabolite profiles. By analyzing metabolites, insights into the physiological state of microorganisms relevant to soil ecosystem phenotypes can be obtained [11], reflecting the interactions between microorganisms and their environment, which assists in evaluating the soil ecosystem’s health [12]. For example, studies have identified soil metabolic markers in tobacco fields affected by green wilt disease by comparing metabolite composition between diseased and healthy soils [13].

Phosphate-solubilizing bacteria, a key category of microbial fertilizers, are increasingly being studied for their role in regulating soil phosphorus cycling and their interactions with microbial communities. The genera *Bacillus*, *Burkholderia*, *Escherichia*, and *Bradyrhizobium* are notable for their phosphate-solubilizing capabilities [14,15]. Research indicates that inoculating soils with phosphate-solubilizing bacteria enhances the activity of phosphorus-related enzymes, thereby promoting phosphorus cycling [16]. These microorganisms facilitate the release of insoluble phosphorus compounds in the soil by secreting metabolites or collaborating with other microbes, making phosphorus more accessible for plant uptake and ultimately enhancing plant growth [17,18]. Furthermore, phosphate-solubilizing bacteria can indirectly promote soil ecosystem health and stability by modulating microbial community structure and function [19].

This study aims to explore the effects of inoculating phosphate-solubilizing bacteria on the soil microbial community of *T. grandis* seedlings, and how these microbial influences on soil metabolic pathways indirectly affect soil phosphorus cycling, thereby enhancing the availability of soil phosphorus. A highly phosphate-solubilizing strain of Burkholderia was inoculated into the soil of *T. grandis* potted seedlings. The strains originated from previous research conducted in the laboratory [20]. By comparing the physicochemical properties of rhizosphere and bulk soils between control and treated groups, this study analyzed bacterial and fungal diversity and community composition, alongside soil metabolite differences. Correlation analysis was then used to explore the associations between differential microbial species and metabolites, uncovering the relationship between microbial community shifts induced by phosphate-solubilizing bacterial inoculation and changes in soil metabolites. The findings of this study provide valuable insights for the effective enhancement of soil quality in *T. grandis* plantations and the long-term sustainability of *T. grandis*.

## 2. Results

### 2.1. Soil Physicochemical Properties of Different Groups

The analysis of soil physicochemical properties demonstrates that inoculation with phosphate-solubilizing bacteria significantly enhanced the total phosphorus, potassium, and nitrogen levels in the rhizosphere soil (RS) of *T. grandis* potted seedlings (*p* < 0.05) (Figure 1). Additionally, it notably increased the available phosphorus, total phosphorus, potassium, and nitrogen concentrations in the bulk soil (BS), with even greater statistical significance (*p* < 0.001).

### 2.2. Changes in Soil Microbial Diversity in Different Groups

We studied the changes in microbial community structure through amplicon sequencing. ASV information for all bacterial species and all fungal species are shown in Appendix A. At the phylum level, the treatment group showed a decrease in Proteobacteria and an increase in Acidobacteria in the rhizosphere soil, while Proteobacteria decreased and Chloroflexi increased in the bulk soil. In the bulk soil, Ascomycota, Basidiomycota, and Chytridiomycota numbers declined, whereas in the rhizosphere soil, Ascomycota decreased significantly and Basidiomycota exhibited a notable increase (Figure 2a,b). Alpha diversity, commonly used to assess species diversity in localized, uniform habitats, is typically measured using indices such as Chao1, Shannon, and Simpson. The Shannon index, in particular, reflects species richness, with higher absolute values indicating greater diversity within the habitat. Box plots display the variations in alpha diversity indices across different groups (Figure 2c,d). The results indicate that after the application of the bacterial agent, the Shannon and Simpson indices of the fungal community declined, while those of the bacterial community rose, suggesting an overall increase in bacterial diversity in both rhizosphere and bulk soils. Principal component analysis (PCA) results reveal that PC1 and PC2 explain 22.8% and 47.4% of the variation in bacterial species, respectively, with a cumulative contribution of 70.2% (Figure 2e,f). For fungal species, PC1 accounts for 84.2% of the variation, and PC2 explains 6.7%, totaling a cumulative contribution of 90.9%. The distinct group separations indicate significant structural differences in different groups.

### 2.3. Differential Microbial Species in Different Groups

Utilizing the metagenomeSeq method on microbial ASVs derived from amplicon sequencing data enabled differential abundance analysis through pairwise comparisons between treatment and control groups. The analysis identified distinct bacterial community shifts in bulk soil, specifically in Acidobacteria, Actinobacteria, Bacteroidetes, Chloroflexi, and Proteobacteria (Figure 3a), while the fungal community exhibited significant variation in the genus Penicillium within Ascomycota (Figure 3b). In the rhizosphere soil, the bacterial communities showing differential abundance included Acidobacteria, Chloroflexi, and Patescibacteria (Figure 3c), with notable fungal differences observed in the phyla Ascomycota and Basidiomycota (Figure 3d).

### 2.4. Enrichment Analysis of Bacterial and Fungal KEGG Metabolic Pathways

The metabolic pathways potentially implicated in the soil microbiota of *T. grandis* were evaluated through KEGG annotation. A comparative analysis between the treated and control groups revealed that certain significantly upregulated and downregulated metabolic pathways for fungi and bacteria. In bulk soil, bacterial pathways such as chorismate biosynthesis II (archaea), aerobic actinobin biosynthesis, sucrose biosynthesis III, and factor 420 biosynthesis were significantly upregulated, while the 3-hydroxypropanoate cycle, glyoxylate assimilation, starch degradation III, and phospholipases were downregulated (Figure 4a). In rhizosphere soil, bacterial pathways involved in dTDP-N-acetylviosamine biosynthesis, benzoyl-CoA degradation II (anaerobic), glycolysis V (Pyrococcus), and factor 420 biosynthesis were upregulated, whereas the superpathway of L-tryptophan biosynthesis, L-glutamate degradation, mannosylglycerate biosynthesis, and nicotine degradation II (pyrrolidine pathway) were downregulated (Figure 4b). Fungal pathways in bulk soil showed an upregulation in 5-aminoimidazole ribonucleotide biosynthesis I, phosphatidylethanolamine biosynthesis (yeast), and phospholipid remodeling, while L-tryptophan degradation, L-arginine biosynthesis II (acetyl cycle), γ-glutamyl cycle, and gluconeogenesis I were downregulated (Figure 4c). In rhizosphere soil, fungal pathways such as L-arginine biosynthesis II (acetyl cycle), methionine biosynthesis II, ubiquinol-7 biosynthesis (eukaryotic), and ubiquinol biosynthesis (eukaryotic) were upregulated, while 5-aminoimidazole ribonucleotide biosynthesis I and gluconeogenesis I were downregulated (Figure 4d).

### 2.5. Screening and Content Changes of Differential Metabolites in Different Groups

Through high-resolution untargeted metabolomics analysis, a total of 888 soil metabolites were obtained. The detailed data of the soil metabolites are shown in Appendix A. A total of 12 categories of metabolites were identified in the positive-ion mode (POS) and 10 categories of significantly different metabolites were detected in the negative-ion mode (NEG) between the control and treatment groups (Figure 5a). Using the OPLS-DA model, differential metabolites in the soil were screened based on OPLS-DA VIP > 1 and *p* < 0.05, revealing that the main differential metabolites included amino acids, fatty acids, hormones, drugs, and other organic compounds (Figure 5d,e). The inoculation with the bacterial agent resulted in the degradation of certain metabolites in the soil, including a reduction in the concentrations of environmental pollutants like Carboplatin and Pinoxaden, as well as a decrease in amino acid compounds such as Tryptophan-Glutamic, Trp-Glu-Arg, and Cys-Gln. Additionally, an increase in the levels of phospholipid metabolites, such as DPPG and Stearoyl Ethanolamide (MEA), was observed. These changes in metabolites may be closely linked to shifts in soil microbial abundance.

### 2.6. Correlation Analysis Reveals the Close Relationship Between Soil Bacteria Communities, Fungi Communities, and Metabolites

Correlation analysis was performed to preliminarily identify soil microbial species associated with changes in soil metabolites (Figure 6). In bulk soil (Figure 6a), the abundance of Chloroflexi showed significant positive correlations with metabolites such as Cys-Gln, Cilastatin, FA 18:2+10, MEA, 21-hydroxyprogesterone, and DPPG (*p* < 0.05). Additionally, Patescibacteria, Planctomycetes, and Cyanobacteria were significantly correlated with certain metabolites (*p* < 0.05). For fungi in bulk soil (Figure 6b), Ascomycota exhibited significant negative correlations with metabolites like 1-oleoyl-2-hydroxy-sn-glycero-3-phospho-(1′-rac-glycerol), Cilastatin, FA 18:2+20, 21-hydroxyprogesterone, and MEA (*p* < 0.05).

In rhizosphere soil (Figure 6c), significant correlations were observed between bacterial communities and differential metabolites. The phyla Proteobacteria, Chloroflexi, Planctomycetes, and Acidobacteria displayed significant correlations with most metabolites (*p* < 0.05). Proteobacteria showed positive correlations with metabolites such as Trp-Glu-Arg, Lys-Met-Lys, 4-hydroxynonenal glutathione, and N6-(1-iminoethyl)-L-lysine (*p* < 0.05), while Chloroflexi, Planctomycetes, and Acidobacteria were negatively correlated with metabolites like 4-hydroxynonenal glutathione, Lys-Met-Lys, and Trp-Glu-Arg. Notably, Acidobacteria demonstrated a positive correlation with 2′-deoxyuridine 5′-monophosphate (*p* < 0.05). Verrucomicrobia and Acidobacteria were significantly positively correlated with metabolites such as Zearalenone and Maltol (*p* < 0.05). For fungi in the rhizosphere (Figure 6d), Ascomycota showed significant positive correlations with Lys-Met-Lys, Trp-Glu-Arg, N6-(1-iminoethyl)-L-lysine, and 4-hydroxynonenal glutathione (*p* < 0.05), while Mucoromycota was positively correlated with 4-hydroxynonenal glutathione, 1,3-propanediol, and Bisphenol A diglycidyl ether (*p* < 0.05).

These results suggest that bacterial phyla such as Chloroflexi, Acidobacteria, Patescibacteria, Planctomycetes, Cyanobacteria, and fungal phyla including Ascomycota and Mucoromycota play key roles in shaping the soil’s metabolite composition.

## 3. Discussion

### 3.1. Impact of Phosphate-Solubilizing Bacterial Agents on Soil Nutrients and Microbial Diversity

Phosphate-solubilizing microorganisms, employed as biofertilizers, have demonstrated significant potential in enhancing soil phosphorus availability [21]. These phosphorus biofertilizers offer an effective approach to sustainable agriculture by linking soil microbial community health with overall soil conditions [22]. Our experiment revealed the effectiveness of phosphate-solubilizing bacterial agents in improving soil nutrient content, further validating their potential in agricultural soil management. Enhancing nutrient levels not only fosters healthy plant growth but also reduces the reliance on chemical fertilizers, thereby minimizing environmental impacts. This contributes significantly to achieving sustainable agriculture. This study employed amplicon sequence variant classification and statistical analysis to assess microbial diversity, examining the effect of bacterial agents on soil microbial community structures. Our findings indicate that the predominant bacterial communities in the soil consisted of Proteobacteria, Actinobacteria, Acidobacteria, Bacteroidetes, and Chloroflexi, aligning with the research of Jiang Niwen (2022) and Zhu Wenjuan (2023) [23,24]. Further analysis showed that bacterial agent application significantly altered the relative abundance of dominant bacteria and fungi. In the rhizosphere soil, the population of Acidobacteria increased, consistent with the findings of Yang Anna et al. (2019) regarding the pivotal role of Acidobacteria in constructing soil ecosystems [25]. Additionally, phosphate-solubilizing bacteria are thought to enhance Acidobacteria enrichment by secreting organic acids that facilitate the transformation and absorption of insoluble phosphates [26]. In bulk soil, the observed rise in Chloroflexi may be attributed to their essential role in soil ecosystems, particularly in carbon and energy cycles [27]. The dominant fungal communities, primarily Ascomycota and Basidiomycota, mirrored the fungal diversity observed in *T. grandis* roots, as noted by Wang Chen et al. (2023) [28]. The sharp decline in Ascomycota and the corresponding increase in Basidiomycota in the rhizosphere soil could stem from the competitive dynamics between these two fungal groups [29]. Additionally, the relatively lower richness and species diversity of fungi compared to bacteria may be due to the latter’s better adaptability to environmental fluctuations [30].

In summary, phosphate-solubilizing microorganisms not only boost nitrogen, phosphorus, and potassium content in *T. grandis* seedling soil but also substantially modify the soil microbial community structure, thereby positively influencing soil health and plant growth. Future studies should focus on exploring the long-term effects of these microbial community shifts on soil ecosystem functions and further investigating how phosphate-solubilizing microorganisms can be harnessed to promote sustainable agriculture.

### 3.2. Functional Prediction of Bacterial and Fungal Communities

Microbial functional genes and metabolic processes play a critical role in the biosynthesis of natural products, cellular activities, and various other metabolic pathways. Functional analysis of microbial communities provides valuable insights into the mechanisms behind soil phosphorus cycling [31]. Utilizing PICRUSt2 for predictive functional analysis, it was found that microbial communities participate in numerous metabolic pathways integral to phosphorus metabolism, forming a complex network that regulates soil phosphorus cycling. The introduction of phosphorus-solubilizing bacterial agents alters these metabolic processes, indirectly enhancing the efficient use and recycling of soil phosphorus.

In bulk soil, bacterial pathways such as chorismate biosynthesis II (potentially linked to archaea), sucrose biosynthesis, and factor 420 biosynthesis were upregulated, while starch degradation III and phospholipase pathways were downregulated. Branched-chain organic acids, including shikimic acid, have the ability to bind with soil minerals, promoting phosphorus release [32]. Phospholipases, responsible for hydrolyzing phospholipids, play a pivotal role in phospholipid remodeling, a process involving changes in the composition of cell membrane phospholipids, which may affect microbial phosphorus uptake and utilization [33,34]. The *phoD* gene encodes alkaline phosphatase, which is involved in the mineralization process of phosphorus. Additionally, the *phoC* genes encode acid phosphatases that effectively mineralize organic phosphorus present in the soil [35]. The upregulation of sucrose biosynthesis may be linked to enhanced plant photosynthesis, while the downregulation of starch degradation could indicate reduced microbial demand for carbon and energy or a shift toward alternative carbon sources. In rhizosphere soil, pathways such as glycolysis, factor 420 biosynthesis, and the superpathway of L-tryptophan metabolism were upregulated, whereas L-tryptophan biosynthesis and L-glutamate degradation pathways were downregulated. The increase in glycolysis, a key component of cellular respiration, suggests heightened microbial metabolic activity. Amino acids, such as glutamate, that are secreted by plant roots can modulate microbial activity, playing an essential role in phosphorus cycling and bioavailability [36].

Fungal metabolic pathways also exhibited significant changes. In bulk soil, phospholipid remodeling was upregulated. Phospholipid remodeling represents a pivotal bio-chemical pathway by which cells modulate the lipidic constituents of their membranes. Within the context of phospholipid metabolism, this process is not only instrumental in preserving the structural integrity and functionality of the cellular membrane but also plays a significant role in a multitude of biological processes, including signal transduction and the transport of substances. Phosphatidylcholine transfer protein (CEPT) is primarily involved in the synthesis of phosphatidylethanolamine, but it also plays a role in phospholipid remodeling. By regulating the transport and metabolism of phospholipids, CEPT influences the phospholipid remodeling process [37]. This study provides an initial exploration of these interactions, illustrating how microbial communities regulate soil phosphorus dynamics by modulating specific metabolic pathways. These findings contribute to a deeper understanding of microbial mechanisms in phosphorus cycling, optimizing the use of phosphorus fertilizers, and developing sustainable soil management strategies. However, the predictive accuracy of PICRUSt2 is influenced by sample quality and sequencing depth, posing a limitation to the study. Future research should focus on experimental validation of these predictions and investigate how environmental factors influence microbial function and phosphorus cycling.

### 3.3. Differential Metabolites and Content Changes

Soil metabolites, as direct outputs of plant root exudates and microbial activities, are integral to the functioning of soil ecosystems [38]. Soil metabolomics provides a novel perspective for understanding these ecosystems by revealing how microbial activities and environmental conditions influence soil metabolite composition [39]. In this study, metabolomic analysis highlighted changes in metabolites such as fatty acids, hormones, amino acids, drugs, and other organic compounds, reflecting shifts in microbial community structure and function. Fatty acids, essential components of microbial cell membranes and critical signal molecules, play a significant role in cell growth, differentiation, and assessing microbial life activities [40].

The inoculation of phosphate-solubilizing bacterial agents resulted in a decrease in soil metabolites such as aleuretic acid, tryptophan-glutamic acid-arginine, cysteine-glutamine, carboplatin, and the herbicide pinoxaden. Aleuretic acid, a primary component of lacquer resin with applications in the fragrance industry [41], showed reduced content, potentially reflecting microbial community responses to environmental changes. The decline in amino acids like tryptophan–glutamic acid–arginine and cysteine–glutamine suggests shifts in microbial metabolism, as amino acids are vital for microbial growth and metabolic processes [42]. Additionally, the reduced presence of environmental pollutants such as carboplatin and pinoxaden could be linked to microbial degradation activities, demonstrating the role of microbes in mitigating environmental contamination [43,44]. Conversely, the increase in metabolites like 1,2-Dipalmitoyl-sn-glycero-3-phospho-(1′-rac-glycerol) (DPPG) and stearoyl ethanolamide may be attributed to heightened microbial metabolic activities. DPPG, a phospholipid, indicates potential renewal of cell membranes or shifts in microbial community structure [45]. Phosphate-solubilizing bacteria not only alter the composition and quantity of soil metabolites but also influence the structure and function of soil microbial communities, which are closely linked to soil phosphorus cycling, nutrient availability, and ecosystem health [46,47].

These findings are vital for understanding and managing soil ecosystems, particularly in ecologically sensitive environments like the *T. grandis* “Merrillii” forests, contributing to broader efforts to enhance ecological sustainability.

### 3.4. Correlation Analysis of Differential Metabolites with Soil Microbes

Soil microbial communities play a critical role in the biogeochemical cycling of phosphorus through various metabolic pathways and ecological functions. This study employed correlation analysis to uncover the relationships between microbial communities and soil metabolites, shedding light on how these microbes contribute to soil phosphorus metabolism. The results indicate that the abundance of Chloroflexi was positively correlated with phospholipid compounds such as DPPG and monooleoyl glycerophosphoethanolamine, suggesting their potential role in facilitating phosphorus release and enhancing plant absorption by renewing cell membranes. This implies that promoting Chloroflexi in soil could increase phosphorus availability. The positive correlation between Proteobacteria and amino acid metabolites highlights their involvement in the nitrogen–phosphorus cycle. Amino acid metabolism not only supplies nitrogen to plants but also enhances phosphorus solubility by complexing with metal ions, improving its bioavailability. Similarly, Acidobacteria exhibited a positive correlation with 2′-deoxyuridine 5′-monophosphate, a nucleic acid metabolite, suggesting their contribution to energy conversion and microbial growth, thereby accelerating the phosphorus cycling process. The association of Verrucomicrobia with specific organic compounds indicates their role in the decomposition of organic matter, which may affect soil pH and redox conditions, thereby influencing phosphorus availability. The positive correlation between Mucoromycota and antioxidant metabolites suggests these fungi help preserve microbial activity under oxidative stress, supporting the phosphorus cycle. In contrast, the Ascomycota showed a significant negative correlation with phospholipid compounds such as 1,2-Dipalmitoyl-sn-glycero-3-phospho(1′-rac-glycerol) (DPPG), while the Basidiomycota exhibited a positive correlation with the compound DPPG. This suggests that the Ascomycota may not be as efficient in acquiring and utilizing phosphorus, particularly when it is present in the form of phospholipid compounds. This could potentially lead to an increase in the population of Basidiomycota and a decrease in the population of Ascomycota. These insights highlight the ecological significance of these microbial groups in soil phosphorus cycling. By leveraging these findings, soil management practices can be developed or optimized, such as introducing bacterial agents containing these key microbial taxa, to enhance phosphorus utilization, minimize phosphorus fixation and loss, and promote sustainable agricultural development.

## 4. Materials and Methods

### 4.1. Materials

A strain of *Burkholderia* with a high phosphorus-solubilizing capacity was used in this experiment, with one-year-old *T. grandis* “Merrillii” potted seedlings serving as the test subjects. In this study, W74, previously characterized in our lab, was inoculated into the soil of *T. grandis* potted seedlings to evaluate its potential effects on plant growth and soil improvement [20].

### 4.2. Experimental Design

In the treatment group, the activated *Burkholderia* was inoculated into 50 mL of sterilized LB liquid medium, while the control group received 5 mL of sterile water in 50 mL of sterilized LB medium (Figure 7). Both groups were incubated at 28 °C with shaking at 220 r·min^−1^ for 5 days. The culture was then centrifuged at 12,000 r·min^−1^ for 10 min, and the bacterial pellet was resuspended in sterile water to achieve a concentration of 1 × 10^8^ CFU·mL^−1^. The bacterial suspension was applied to the seedlings’ root zone through drip irrigation, with each pot receiving 10 mL of the suspension every 14 days. The control group received 10 mL of sterile water. Six replicates were used per treatment. After 60 days of cultivation in a constant-temperature chamber set at 26 °C, relative humidity 60–80%, soil samples were collected. The root systems of the *T. grandis* seedlings were gently removed, and the rhizosphere soil (RS) was collected by gently brushing off the soil attached to the roots. Bulk soil (BS) was gathered with a sterilized soil spoon, placed into sterile bags, and mixed thoroughly to form a composite sample. Each soil sample was divided into two portions: one was immediately preserved on dry ice to maintain microbial activity, while the other was stored at −80 °C for long-term preservation.

### 4.3. Physicochemical Analysis of Soil Samples

The collected soil samples were sieved through a 2 mm mesh. The digestion process was conducted using the sulfuric acid-peroxide method, and the total nitrogen, phosphorus, and potassium contents were determined using Inductively Coupled Plasma Mass Spectrometry (ICP-MS) (Icap RQ, Thermofisher, Bremen, Germany). Available phosphorus was assessed with the ammonium fluoride–molybdate blue colorimetric method. Data compilation was performed using Excel 2010, and statistical analysis, including one-way ANOVA and multiple comparisons, was conducted with GraphPad.Prism.9.5.

### 4.4. Soil DNA Extraction, Amplicon Sequencing, and Data Analysis

Following thawing at 4 °C, 0.5 g of soil was collected from each group for DNA extraction using the FastDNA Spin Kit for Soil (code:116560200, MP Biomedicals, Santa Ana, CA, USA). The V3-V4 region of bacterial 16S rDNA was amplified using primers 338F (5′-ACTCCTACGGGAGGCAGCA-3′) and 806R (5′-GGACTACHVGGGTWTCTAAT-3′), while fungal ITS1 amplification employed primers ITS5 (5′-GGAAGTAAAAGTCGTAACAAGG-3′) and ITS2 (5′-GCTGCGTTCTTCATCGATGC-3′). Product concentration and quality were verified through 2% agarose gel electrophoresis and using Qubit and Nanodrop (ThermoFisher, Waltham, MA, USA) before constructing libraries. After qualification, sequencing was performed on the Illumina NovaSeq6000 platform (Illumina, CA, USA).

In QIIME2, the DADA2 method was used to process raw sequences, generating ASV sequences and tables. Bacterial 16S rDNA sequences were compared against the SILVA v13.2 database, and fungal ITS sequences were aligned with the UNITE v8.0 database. Taxonomic classification utilized the classify-sklearn function in QIIME2. Alpha diversity indices, including the Shannon, Simpson, Chao1, and Good’s coverage indices, were computed in QIIME2, and box plots were created using R software (4.2.0). Rarefaction curves were generated with QIIME2 View using the alpha-rarefaction.qzv file. Abundance rank curves for each group were plotted in R3.6.1.

For species composition analysis, the non-rarefied ASV table was used to compare the treatment and control groups, and the metagenomeSeq package (v1.32.0)was employed to identify differences in species composition, providing taxonomic position and statistical significance of the target ASVs. Results were visualized with a Manhattan plot.

### 4.5. Soil Untargeted Metabolomics Analysis and Data Analysis

Metabolomics analysis used the same soil samples as biodiversity analysis. Thawed samples were mixed with a pre-cooled methanol/acetonitrile/water solution (2:2:1, *v*/*v*), vortexed, ultrasonicated at low temperature for 30 min, left at −20 °C for 10 min, and centrifuged at 14,000 r·min^−1^ at 4 °C for 20 min. The supernatant was vacuum-dried, and the residue was resuspended in 100 μL of acetonitrile/water (1:1, *v*/*v*), vortexed, and centrifuged again at 14,000 r·min^−1^ for 15 min. Separation was performed using an Agilent 1290 Infinity LC ultra-high-performance liquid chromatography system, with samples maintained at 4 °C throughout the process. Quality control samples were integrated to ensure result reliability. Mass spectrometry analysis was conducted on a TripleTOF 6600 mass spectrometer(Applied Biosystems, Waltham, MA, USA) using both positive and negative electrospray ionization (ESI) modes. XCMS software (XCMS 2.x) was utilized for peak alignment, retention time correction, and peak area extraction.

## 5. Conclusions

This study investigated the effects of inoculating the phosphate-solubilizing bacterium *Burkholderia* on the soil microbial community and metabolite profile of *T. grandis* seedlings. Results showed increased phosphorus availability, total nutrient content, and significant changes in soil microbial structure, with notable increases in Acidobacteria, Chloroflexi, and Proteobacteria, and shifts in Ascomycota and Basidiomycota. These changes demonstrate benefits to microbial diversity and nutrient cycling. Metabolomic analysis revealed shifts in fatty acids, hormones, amino acids, and other organic molecules, linked to microbial community alterations. Bacteria influenced soil phosphorus metabolism via microbial-driven metabolic pathways. Inoculating with phosphate-solubilizing bacteria is promising for *T. grandis* forestry sustainability, improving phosphorus utilization and soil health. Future research should explore bacterial–microbial interactions, responses to soil management, and environmental conditions for precise nutrient management and ecological balance. Furthermore, the practical implications of these findings extend beyond the cultivation of *T. grandis*, offering potential applications for a broader agricultural context. By optimizing microbial inoculation strategies, the phosphorus use efficiency of other crops can be enhanced, reducing the reliance on chemical fertilizers and thus promoting sustainable agriculture. Future research should further investigate the interactions between bacteria and other microorganisms, as well as their responses to soil management practices and environmental conditions, which will contribute to more precise nutrient management and ecological balance.

## Figures and Tables

**Figure 1 plants-13-03209-f001:**
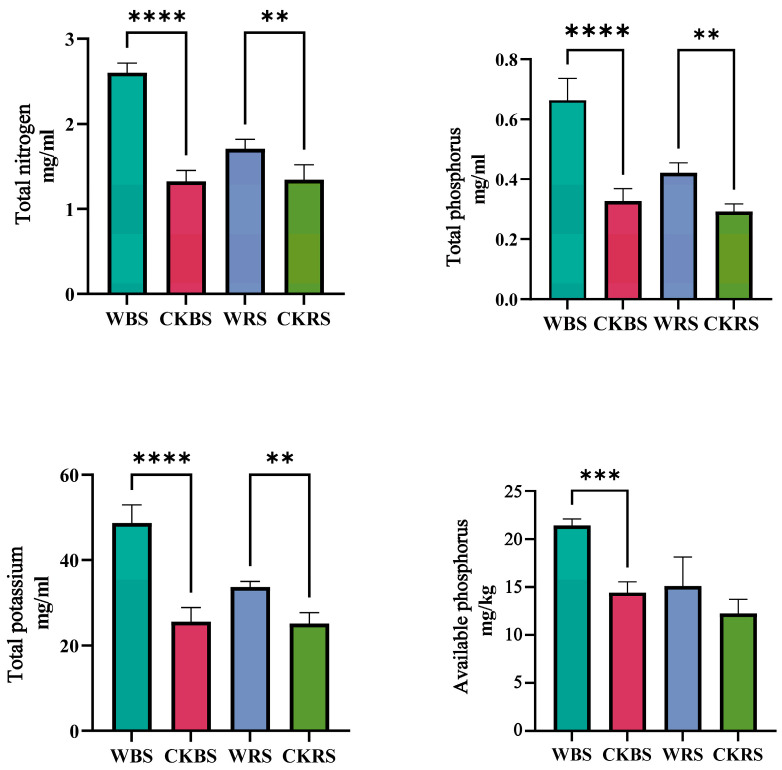
Soil physicochemical properties across the various groups, with statistical significance denoted as * *p* ≤ 0.05, ** *p* ≤ 0.01, and *** *p* ≤ 0.001, **** *p* ≤ 0.0001. “W” signifies the bacterial agent treatment group, “CK” refers to the control group, and “RS” and “BS” represent rhizosphere soil and bulk soil, respectively.

**Figure 2 plants-13-03209-f002:**
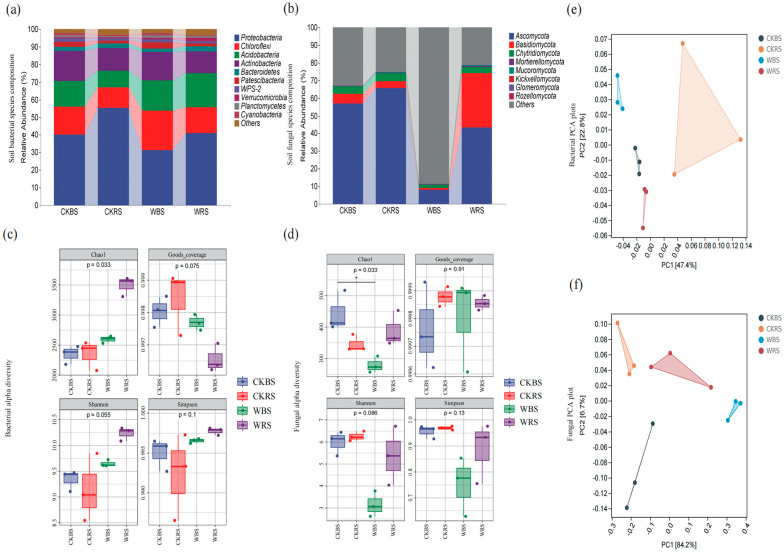
Changes in microbial diversity following treatment with phosphate-solubilizing bacteria in *T. grandis*. Panels (**a**,**b**) show the composition of bacterial and fungal communities, respectively, while (**c**,**d**) display the alpha diversity of these communities. Panels (**e**,**f**) present the principal component analysis (PCA) plots for bacteria and fungi, respectively. In Figures (**a**,**b**), the horizontal axis represents the four subgroups (CKBS, CKRS, WBS, WRS), and the vertical axis reflects the abundance of species at the genus level. The length of the bars corresponds to species abundance. The box plots in Figures (**c**,**d**) depict Chao1, which indicates total species richness, while the Simpson and Shannon indices measure microbial diversity within the soil samples. Good’s coverage reflects sample coverage. The *p*-value atop the box plot indicates the statistical significance of diversity index differences among the subgroups.

**Figure 3 plants-13-03209-f003:**
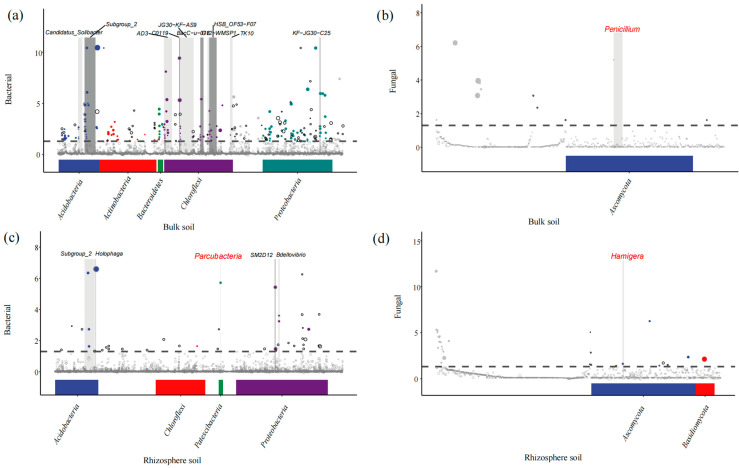
Species-level changes in soil microorganisms across different groups. (**a**) Bacterial changes in the bulk soil. (**b**) Fungal changes in the bulk soil. (**c**) Bacterial changes in the rhizosphere soil. (**d**) Fungal changes in the rhizosphere soil. The horizontal axis represents the ASVs, organized according to their taxonomic information from phylum to species, while the vertical axis reflects the −log10(adj-Pvalue) values. Each dot or circle represents an ASV, with its size indicating relative abundance, expressed in log2(CPM/n). The dotted line represents significance analysis. Microorganisms below the dotted line are not significant. The symbols above the dotted line are significant. A grayscale background highlights the top 10 genera with the most significantly upregulated points (default setting), and the genus name is labeled at the top of the figure.

**Figure 4 plants-13-03209-f004:**
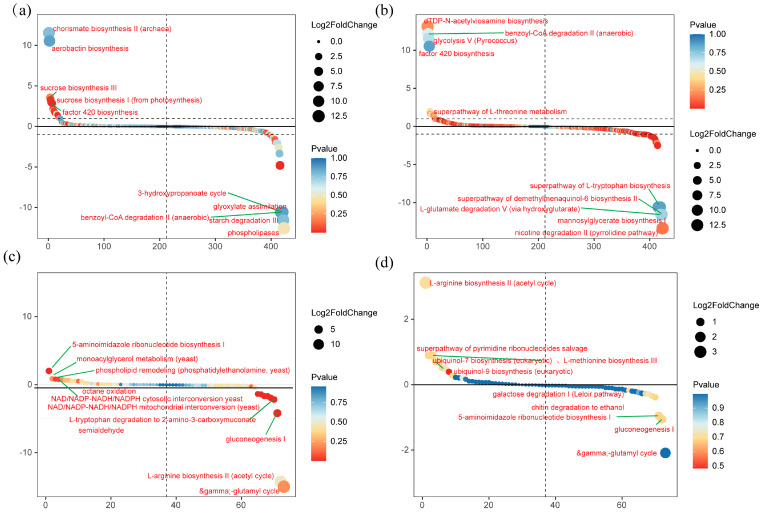
Microbial functional prediction analysis. Functional predictions for bacterial communities in bulk (**a**) and rhizosphere (**b**) soils, and fungal communities in bulk (**c**) and rhizosphere (**d**) soils. The vertical axis reflects the degree of upregulation (positive) or downregulation (negative) following inoculation. The colored dots indicate the significance of pathway changes (represented as the −log10 of the *p*-value), while the size of the dots corresponds to the extent of fold changes (represented as the log2 of the fold change).

**Figure 5 plants-13-03209-f005:**
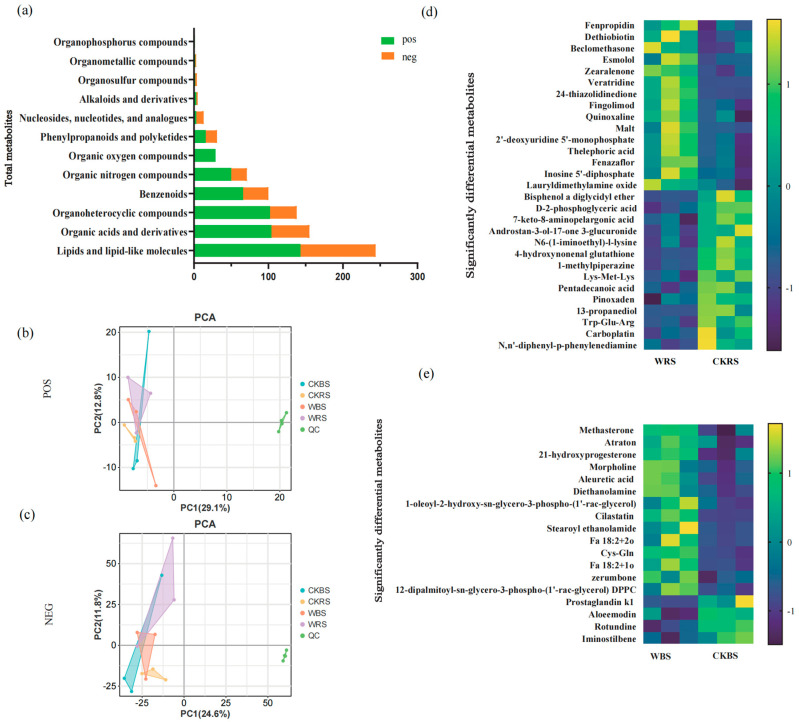
Analysis of significantly differential metabolites in soil. (**a**) Total metabolites identified across all soil samples. PCA of metabolites in positive-ion mode (POS) (**b**) and negative-ion mode (NEG). (**c**) The clustering of three samples in the same group indicates good reproducibility. A heatmap displaying significant differences in soil metabolites for POS (**d**) and NEG (**e**). The vertical axis lists the significantly different metabolites, while the horizontal axis shows the different groups. Yellow indicates elevated expression levels of significantly different metabolites, and blue indicates reduced expression levels. WBS and WRS represent treatment groups, while CKBS and CKRS serve as control groups.

**Figure 6 plants-13-03209-f006:**
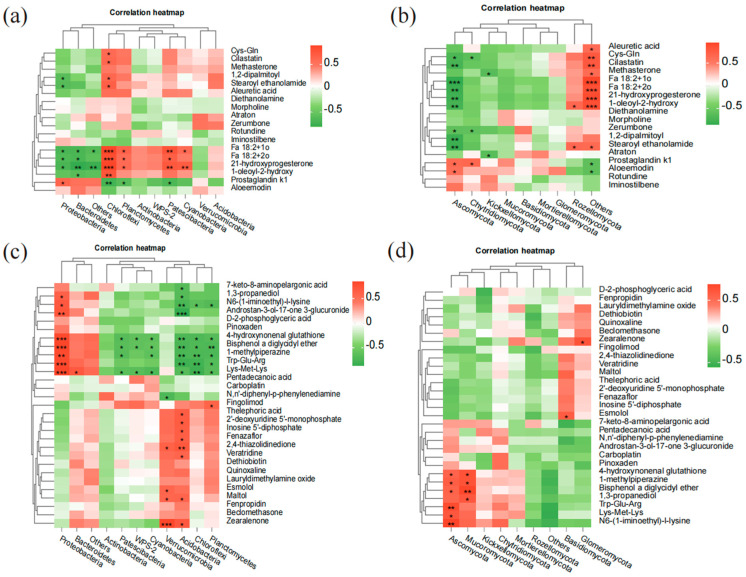
Correlation analysis of bacterial and fungal phyla with significantly differential metabolites.The asterisks (*, **, ***) represent the level of significance, where * indicates *p* ≤ 0.05, ** indicates *p* ≤ 0.01, and *** indicates *p* ≤ 0.001. (**a**) Correlation between bacterial phyla and significantly differential metabolites in bulk soil. (**b**) Correlation between fungal phyla and significantly differential metabolites in bulk soil. (**c**) Correlation between bacterial phyla and significantly differential metabolites in rhizosphere soil. (**d**) Correlation between fungal phyla and significantly differential metabolites in rhizosphere soil. Horizontal axis displays significantly differential metabolites (with some shown as abbreviations), while vertical axis represents microorganisms (ten most abundant phyla). Red indicates positive correlation, and green indicates negative correlation. 1,2-Dipalmitoyl-sn-glycero-3-phospho(1′-rac-glycerol) (DPPG).

**Figure 7 plants-13-03209-f007:**
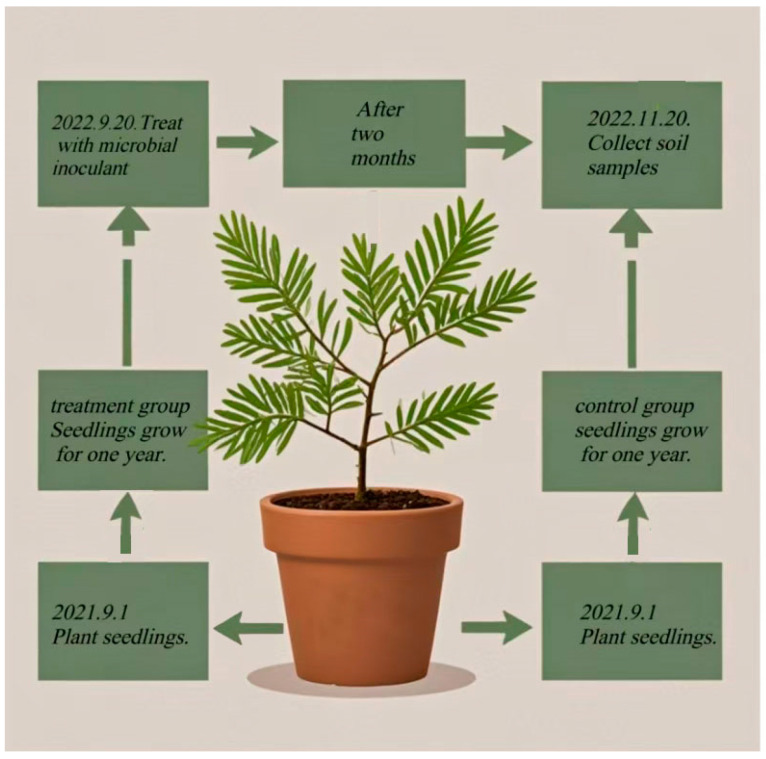
Schematic diagram of pot experiment.

## Data Availability

All data generated or analyzed during this study are included in this published article and its Appendix A.

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
