# Peer review of "Soil Microbial and Metabolomic Shifts Induced by Phosphate-Solubilizing Bacterial Inoculation in *Torreya grandis* Seedlings"

_plants, 2024, doi:10.3390/plants13223209_

Round 1
Reviewer 1 Report
Comments and Suggestions for Authors
The article titled “Effects of Phosphate-Solubilizing Bacterial Inoculation on Soil Microbial Community and Metabolites in Torreya grandis Seedlings” addresses a critical aspect of sustainable agriculture by evaluating the impact of phosphate-solubilizing bacteria (PSB) on nutrient cycling and soil health. The study is significant as it highlights how the inoculation of PSB enhances phosphorus availability, improves microbial diversity, and modulates soil metabolite composition. These findings are vital for promoting the efficient use of fertilizers, reducing environmental pollution, and fostering the long-term sustainability of T. grandis forestry. The results also open up new avenues for optimizing phosphorus management in agricultural systems by leveraging microbial interactions to enhance soil quality and crop productivity. I believe that the manuscript needs Minor revisions.
Minor Comments for the Authors
- Title: Consider rephrasing the title to reflect more precisely the microbial and metabolic impacts of the bacterial inoculation. Including terms like “Microbial and Metabolomic Shifts” could enhance clarity and emphasize the focus on microbial diversity and metabolite changes.
- Abstract: The abstract is well-written but can be condensed to emphasize the main findings and reduce the introduction of background information. A more concise abstract will improve readability and quickly convey the novelty of the study.
- Introduction: The introduction provides relevant background, but the transitions between the ecological and biochemical significance of phosphorus could be smoother. Consider reorganizing the paragraphs to ensure a logical flow from the problem (phosphorus availability) to the solution (phosphate-solubilizing bacteria) and the study's objectives.
- Figure Legends: Some figure legends, such as for Figure 5, could benefit from additional clarity. Explain key terms like POS and NEG and ensure the legends are fully understandable without referencing the main text.
- Discussion: The discussion is thorough, but in places, it can be condensed to avoid redundancy. Focus on synthesizing the key points rather than reiterating results in detail. Also, consider expanding on the potential long-term impacts of microbial shifts on soil health, linking it back to agricultural sustainability.
- Conclusions: The conclusions could further highlight the practical implications of the findings, especially how they can be applied in broader agricultural contexts beyond T. grandis. Mentioning future research directions in microbial interactions and nutrient cycling would strengthen the conclusion.
- References: Ensure all references follow the same format and include DOIs or URLs where applicable. There are a few discrepancies in how journal names and authors are formatted.
- Supplementary Information: While the supplementary data is referenced, a brief explanation of its content and significance in the results section would provide additional clarity for readers.
The English could be improved to more clearly express the research and check there are several grammatical mistakes throughout.
Author Response
Comment1. Title: Consider rephrasing the title to reflect more precisely the microbial and metabolic impacts of the bacterial inoculation. Including terms like “Microbial and Metabolomic Shifts” could enhance clarity and emphasize the focus on microbial diversity and metabolite changes.
Response1: First and foremost, we would like to express our sincere gratitude for the time and effort you have dedicated to reviewing our manuscript. Your insights and suggestions are invaluable to us, as they have helped us to view our research from different perspectives and have significantly enhanced the quality of our work. We have revised the title to "Soil Microbial and Metabolomic Shifts Induced by Phosphate-Solubilizing Bacterial Inoculation in Torreya grandis Seedlings" to better capture the essence of our research.
Comment2. Abstract: The abstract is well-written but can be condensed to emphasize the main findings and reduce the introduction of background information. A more concise abstract will improve readability and quickly convey the novelty of the study.
Response2:Thank you for your valuable feedback on our abstract. We have taken your suggestion to reduce the background information to make the abstract more concise. We have deleted “ Torreya grandis, a member of the Taxaceae family and Torreya genus,it considerable economic significance. ” in Abstract.
Comment3. Introduction: The introduction provides relevant background, but the transitions between the ecological and biochemical significance of phosphorus could be smoother. Consider reorganizing the paragraphs to ensure a logical flow from the problem (phosphorus availability) to the solution (phosphate-solubilizing bacteria) and the study's objectives.
Response3: Thank you for your insightful comments on the introduction section of our manuscript. We appreciate your suggestion to enhance the transitions between the ecological and biochemical significance of phosphorus. To address this, we have reorganized the paragraphs to ensure a more logical progression from the problem of phosphorus availability to the solution involving phosphate-solubilizing bacteria, and finally to the objectives of our study. We have added “This study aims to explore the effects of inoculating phosphate-solubilizing bacteria on the soil microbial community of T.grandis seedlings, and how these microbial influences on soil metabolic pathways indirectly affect soil phosphorus cycling, thereby enhancing the availability of soil phosphorus.”in line 76-79.
Comment4. Figure Legends: Some figure legends, such as for Figure 5, could benefit from additional clarity. Explain key terms like POS and NEG and ensure the legends are fully understandable without referencing the main text.
Response4: Thank you for drawing our attention to the clarity of the figure legends, particularly for Figure 5. We have taken your suggestion and have now included explanations for key terms such as POS and NEG within the figure legends themselves. This ensures that the legends are comprehensive and understandable without the need to reference the main text.Wu have added“The clustering of three samples in the same group indicates good reproducibility”in line 211-212.
Comment5. Discussion: The discussion is thorough, but in places, it can be condensed to avoid redundancy. Focus on synthesizing the key points rather than reiterating results in detail. Also, consider expanding on the potential long-term impacts of microbial shifts on soil health, linking it back to agricultural sustainability.
Response5:We have carefully revised the section, focusing on synthesizing the key points and avoiding unnecessary repetition. We have also expanded on the potential long-term impacts of microbial shifts on soil health and their implications for agricultural sustainability, ensuring that these discussions are well-integrated and provide a broader context to our findings.We have deleted “that the inoculation of phosphate-solubilizing bacterial agents markedly increased the levels of total phosphorus, potassium, and nitrogen in both rhizosphere and bulk soils of T.grandis seedlings, with a significant rise in available phosphorus in bulk soil (P < 0.05). These results emphasize”and added “the effectiveness of phosphate-solubilizing bacterial agents in improving soil nutrient content, further validating their potential in agricultural soil management. Enhancing nutrient levels not only fosters healthy plant growth but also reduces the reliance on chemical fertilizers, thereby minimizing environmental impacts. This contributes significantly to achieving sustainable agriculture” in line 263-267.We have deleted “while L-tryptophan degradation, L-arginine biosynthesis II (acetyl cycle), γ-glutamyl cycle, and gluconeogenesis I were downregulated”and “ In rhizosphere soil, the upregulation of L-arginine biosynthesis II (acetyl cycle) and methionine biosynthesis, alongside the downregulation of 5-aminoimidazole ribonucleotide biosynthesis I and gluconeogenesis I, underscores the complex role fungi play in phosphorus metabolism” in line 332-336.
Comment6. Conclusions: The conclusions could further highlight the practical implications of the findings, especially how they can be applied in broader agricultural contexts beyond T. grandis. Mentioning future research directions in microbial interactions and nutrient cycling would strengthen the conclusion.
Response6: Your comments were valuable. We have revised our conclusions to emphasize the practical significance and potential for future studies on microbial interactions.We have deleted “Furthermore, the practical implications of these findings extend beyond the cultivation of T.grandis, offering potential applications for a broader agricultural context. By optimizing microbial inoculation strategies, the phosphorus use efficiency of other crops can be enhanced, reducing the reliance on chemical fertilizers and thus promoting sustainable agriculture. Future research should further investigate the interactions between bacteria and other microorganisms, as well as their responses to soil management practices and environmental conditions, which will contribute to more precise nutrient management and ecological balance” in line 496-504.
Comment7. References: Ensure all references follow the same format and include DOIs or URLs where applicable. There are a few discrepancies in how journal names and authors are formatted.
Response7: We appreciate your feedback and have corrected the inconsistencies in reference formatting, ensuring all follow the same style.
Comment8. Supplementary Information: While the supplementary data is referenced, a brief explanation of its content and significance in the results section would provide additional clarity for readers.
Response8: Thank you for your suggestion. We have added a brief explanation of the supplementary data in the results section for clarity.We have added “ASV information for all bacterial species and all fungal species are shown in Table S1 and Table S2” and “ The detailed data of the soil metabolites are shown in Table S3”.
Reviewer 2 Report
Comments and Suggestions for Authors
The manuscript describes using phosphate-solubilizing bacteria to improve soil quality and determine the biodiversity of microbiota and metabolites. Overall, the manuscript contains useful information. However, minor points need to be clarified:
Abstract:
- Scientific name of organism. It is the first time in full name, and later on, it should be written in short (genus). Please also check and correct throughout the manuscript.
Keywords:
- Avoid using the same words as in the article topic.
Materials and methods:
- The supplier of the FastDNA Spin Kit for Soil (MP Biomedicals) used in this study should be given, such as the country where this kit is produced or sole.
2% agarose gel electrophoresis can not be utilized to precisely determine the concentration and quality of the gDNA or PCR product. Please elaborate on this point.
Discussion:
- What kind of compounds may affect the growth compatibility between Ascomycota and Basidiomycota?
- Please elaborate more regarding the genes or proteins that may be involved in several biosynthesis pathways or metabolite production during phosphorus metabolism.
Author Response
General comments
The English could be improved to more clearly express the research and check there are several grammatical mistakes throughout.
Response:Thank you very much for your positive evaluation. The comments and suggestions are valuable and very helpful for revising and improving our manuscript, as well as the important guiding significance to our researches. The manuscript has certainly benefited from these insightful revision suggestions. The article has been polished by the Bullet Edits (http://bulletedits.com/). We make every effort to minimize language errors.
Comments and Suggestions for Authors
The manuscript describes using phosphate-solubilizing bacteria to improve soil quality and determine the biodiversity of microbiota and metabolites. Overall, the manuscript contains useful information. However, minor points need to be clarified:
Comment1.Scientific name of organism. It is the first time in full name, and later on, it should be written in short (genus). Please also check and correct throughout the manuscript.
Response1:Thank you for pointing this out. We have abbreviated the biological names that appear for the second time in the full text.
Comment2. Avoid using the same words as in the article topic.
Response2:Thank you for the suggestion. We have revised the keywords to include alternative terms that represent the core content without duplicating the article title.We have modified the keywords to avoid the same as the title.
Comment3.The supplier of the FastDNA Spin Kit for Soil (MP Biomedicals) used in this study should be given, such as the country where this kit is produced or sole.
Response3:Thank you for your inquiry. The supplier of the FastDNA Spin Kit for Soil is MP Biomedicals, with the product being manufactured in the United States, and the catalog number is 116560200 (50 PREPS).We have added “(code:116560200, MP Biomedicals ,USA)” in line 449.
Comment4. 2% agarose gel electrophoresis can not be utilized to precisely determine the concentration and quality of the gDNA or PCR product. Please elaborate on this point.
Response4:Thank you for your keen observation. We have added “and useing Qubit and Nanodrop (ThermoFisher, USA)” in line 455-456.
Comment5. What kind of compounds may affect the growth compatibility between Ascomycota and Basidiomycota?
Response5: In contrast, the Ascomycota showed a significant negative correlation with phospholipid compounds such as 1,2-Dipalmitoyl-sn-glycero-3-phospho(1'-rac-glycerol)(DPPG), while the Basidiomycota exhibited a positive correlation with the compound DPPG. This suggests that the Ascomycota may not be as efficient in acquiring and utilizing phosphorus, particularly when it is present in the form of phospholipid compounds. This could potentially lead to an increase in the population of Basidiomycota and a decrease in the population of Ascomycota. We have deleted “potentially due to their dual function in organic matter decomposition and phosphorus metabolism” and added “the Ascomycota showed a significant negative correlation with phospholipid compounds such as 1,2-Dipalmitoyl-sn-glycero-3-phospho(1'-rac-glycerol) (DPPG), while the Basidiomycota exhibited a positive correlation with the compound DPPG. This suggests that the Ascomycota may not be as efficient in acquiring and utilizing phosphorus, particularly when it is present in the form of phospholipid compounds. This could potentially lead to an increase in the population of Basidiomycota and a decrease in the population of Ascomycota.” in line 400-407.
Comment6. Please elaborate more regarding the genes or proteins that may be involved in several biosynthesis pathways or metabolite production during phosphorus metabolism.
Response6:The phosphatase pathway plays an essential role in phosphorus metabolism, involving the release and recycling of phosphorus, particularly under phosphorus-limited conditions. This pathway is crucial for maintaining intracellular phosphorus balance and the efficient utilization of phosphorus. Phosphatases can degrade phospholipid compounds present in the soil, making them available for microbial use. Under phosphorus limitation, the expression of genes involved in this process is typically upregulated to enhance the cell's ability to acquire and utilize phosphorus. In the phosphatase pathway, the phoD gene encodes another alkaline phosphatase involved in the mineralization of phosphorus. The phoC gene encodes acid phosphatase, which effectively mineralizes organic phosphorus in the soil. The Phosphatidylcholine transfer protein (CEPT) is primarily involved in the synthesis of phosphatidylethanolamine (PE), but it also plays a role in phospholipid remodeling. By regulating the transport and metabolism of phospholipids, CEPT influences the phospholipid remodeling process. Phosphatidylethanolamine (PE) plays a significant role in phosphorus metabolism; it is not only a key structural component of the cell membrane but also participates in various cellular processes and signal transduction. Phospholipid remodeling is an important biochemical process by which cells regulate the composition of their membrane phospholipids, altering the fatty acid chain composition, which in turn affects the biophysical properties and biological functions of the membrane. In phosphorus metabolism, phospholipid remodeling not only maintains the structure and function of the cell membrane but also participates in signal transduction, substance transport, and other biological processes.Thank you very much for your insightful question regarding the genes and proteins involved in phosphorus metabolism, particularly within the context of biosynthesis pathways and metabolite production.
We have added “The phoD gene encodes alkaline phosphatase, which is involved in the mineralization process of phosphorus. Additionally, the phoC genes encode acid phosphatases that effectively mineralize organic phosphorus present in the soil” in line 313-316 and “Phospholipid remodeling represents a pivotal bio-chemical pathway by which cells modulate the lipidic constituents of their membranes. Within the context of phospholipid metabolism, this process is not only instrumental in preserving the structural integrity and functionality of the cellular membrane but also plays a significant role in a multitude of biological processes, including signal transduction and the transport of substances. Phosphatidylcholine transfer protein (CEPT) is primarily involved in the synthesis of phosphatidylethanolamine, but it also plays a role in phospholipid remodeling. By regulating the transport and metabolism of phospholipids, CEPT influences the phospholipid remodeling process” in line 327-336.